# Latent Human Cytomegalovirus Infection Activates the STING Pathway but p-IRF3 Translocation Is Limited

**DOI:** 10.3390/v17081109

**Published:** 2025-08-12

**Authors:** Wang Ka Lee, Zuodong Ye, Allen Ka Loon Cheung

**Affiliations:** Department of Biology, Faculty of Science, Hong Kong Baptist University, Kowloon Tong, Hong Kong SAR, China; 19482124@life.hkbu.edu.hk (W.K.L.); zuodongye2022@163.com (Z.Y.)

**Keywords:** HCMV, latent infection, cell differentiation, innate immune pathway, IRF3

## Abstract

Human cytomegalovirus (HCMV) is a ubiquitous betaherpesvirus that establishes lifelong latent infection in CD34^+^ haematopoietic stem and progenitor cells. A unique subset of viral genes is expressed during latency, which functions to modulate cellular mechanisms without supporting viral replication. One potential function of these genes is to regulate the differentiation state of latently infected CD34^+^ cells, thereby preventing their progression into antigen-presenting cells, e.g., dendritic cells. In this study, we first compared CD34^+^ cells that supported productive and latent infections using the RV-TB40-BAC_KL7_-SE-EGFP virus. Over a seven-day time course, the proportion of latently infected CD34^+^ cell subsets within the myeloid progenitor population remained similar to that in the mock-infected control. However, starting from day 3 post-infection, there was an increase in the proportion of the early progenitor subsets, including haematopoietic stem cells (HSCs) and multipotent progenitors (MPPs). In contrast, productively infected cells, which constituted less than 1% of the population, only accounted for a small portion of the myeloid progenitors. Importantly, our data revealed that the innate immune STING/p-TBK1/p-IRF3 pathway was activated in latently infected CD34^+^ cells, yet type I interferon (IFN) expression was decreased. This decrease was attributed to impaired p-IRF3 nuclear translocation, limiting the induction of an autocrine type I IFN response. However, treatment with IFN-β could induce myelopoiesis in latently infected cells. In summary, HCMV modulates a key component of the STING pathway to inhibit antiviral immune responses by decreasing the type I IFN-mediated cell differentiation of CD34^+^ progenitor cells. This study uncovered a new mechanism of latent HCMV-mediated regulation of the host cell differentiation response.

## 1. Introduction

Human cytomegalovirus (HCMV) is a ubiquitous betaherpesvirus that has infected 60–90% of the global population [1]. The virus establishes a lifelong latent infection in CD34^+^ haematopoietic stem and progenitor cells (HSPCs) and myeloid lineage cells within the human host, typically remaining asymptomatic unless immune surveillance is compromised. This can occur in immunocompromised HIV-1 patients, immunosuppressed transplant recipients, and individuals undergoing intensive chemotherapy [2,3,4,5,6]. However, the persistence of latent HCMV has been identified as a potential factor in the development of certain diseases, including post-transplant organ pathologies such as hepatitis and pneumonia [7], congenital conditions like biliary atresia [8,9], and autoimmune diseases [10]. Moreover, the immune memory inflation phenomenon induced by latent HCMV may potentially impair immune homeostasis in ageing individuals [11,12].

Lytic and latent HCMV infections are distinguished by unique viral gene transcriptional profiles [5,13]. The major immediate-early promoter (MIEP) drives a cascade of viral genes that facilitate lytic/productive infection but is repressed during latent infection, when there is minimal viral gene expression [5,14]. A set of latent genes was identified using microarrays, and more recent single-cell sequencing data revealed that a broad range of viral genes are expressed, albeit at very low levels, during latency [13,15,16,17,18,19,20,21,22,23]. Several latent genes have immunomodulatory functions. For example, UL111.5A, encoding the viral IL-10 homolog (LAcmvIL-10), induces host IL-10 production and downregulates major histocompatibility complex (MHC) class II expression [24,25,26]. UL138 maintains latency by altering tumour necrosis factor (TNF) signalling and inhibiting innate immune pathways [23,27,28,29]. LUNA promotes latency through chromatin remodelling by deSUMOylation of promyelocytic leukaemia protein (PML), a key regulator of viral reactivation [30]. US28 represses MIEP activity to inhibit lytic gene expression and support latency [31,32,33,34]. UL135 and UL136 have also been implicated in the latency–reactivation balance, particularly through endosomal remodelling and immune modulation [35,36]. Overall, the key functions of HCMV latent genes are to suppress immune-related factors, enabling viral evasion of immunosurveillance and maintenance of dormancy within reservoir populations.

Another critical aspect of HCMV latency is the impairment of CD34^+^ HSPC differentiation [5,37]. CD34^+^ HSPCs are capable of generating innate immune responses or becoming antigen-presenting cells, such as dendritic cells, to induce adaptive antiviral immune responses [14,38,39]. An early study showed that HCMV DNA can be detected in both CD34^+^ HLA-DR^−^ and CD34^+^HLA-DR^+^ populations, and that HCMV suppresses proliferation within these populations [40]. More recently, HCMV latency was shown to be established in CD38^+^CD90^+^ and CD38^+/low^CD90^−^ CD34^+^ haematopoietic progenitor cells, while viral persistence was observed in CD38^−^CD90^+^ cells [37]. These findings indicate that the differentiation status and phenotypic identity of CD34^+^ subsets play a critical role in determining susceptibility to HCMV latency.

HCMV may further reinforce latency by interfering with host differentiation pathways. For instance, the latent gene LAcmvIL-10 inhibits pro-inflammatory cytokines, preventing the latently infected myeloid progenitor cells from differentiating into dendritic cells [41]. The latent miRNA miR-US5-2 targets the transcriptional repressor NAB1 and upregulates TGF-β expression, inducing myelosuppression [42,43]. HCMV also regulates the activity of STAT3 to induce HSPC differentiation into an immunosuppressive monocyte subset, which produces high levels of nitric oxide to achieve latency [44]. Together, these strategies allow HCMV to exploit specific progenitor subsets and their developmental plasticity to establish and maintain long-term latency in an immunosuppressive cellular environment.

Type I interferons (IFNs) are triggered by numerous innate sensors and are key players in cellular host defences against virus infection [45,46,47]. Both acute and chronic exposure to type I IFNs can modulate the maintenance of haematopoiesis [45,48,49,50,51]. It has been reported that CD34^+^ cells can undergo IFN-driven haematopoiesis through activation of the cyclic GMP–AMP synthase (cGAS)–STING pathway and autocrine type I IFN signalling [52]. Moreover, recent studies on bacterial infection have shown that c-di-GMP, a bacterial second messenger, binds and activates the STING pathway to regulate HSPC homeostasis and type I IFNs [53]. However, the influence on the expression of type I IFNs and its impact on CD34^+^ HSPCs differentiation during latent HCMV infection are poorly understood. cGAS is a cytosolic DNA sensor that recognises both double-stranded and single-stranded viral DNA and initiates activation of the downstream adaptor protein STING. Indeed, cGAS detects HCMV lytic infection of endothelial cells and triggers a type I IFN response [54]. In addition to cGAS, STING can also be activated by other viral DNA sensors, including DDX41 and IFI16 [55,56,57,58]. Notably, IFI16 is targeted in undifferentiated myeloid cells during latency in a US28-dependent manner [59], suggesting that modulation of innate sensors is involved in latency establishment. Upon activation, STING translocates from the endoplasmic reticulum to the Golgi, where it activates TBK1. TBK1 then phosphorylates IRF3, enabling its nuclear translocation and induction of type I IFN responses [60].

HCMV activates STING in monocyte-derived dendritic cells [61]. As infection progresses, the expression of IFN-β and IFN-stimulated genes is inhibited, further promoting productive infection [61]. Similarly, HCMV has been shown to inhibit TGF-β signalling during lytic infection to limit the induction of type I IFNs [62]. These findings highlight the importance of the STING/TBK1/IRF3 axis and type I IFNs as viral targets for immune evasion and latency establishment [63]. IRF3, a central transcription factor downstream of STING, has been shown to play a pivotal role in HCMV sensing and innate immune signalling [64,65,66,67], yet little is known about how latent infection modulates IRF3 translocation or activity.

In this study, we investigated whether latent HCMV infection modulates the STING/p-TBK1/p-IRF3 pathway to influence CD34^+^ cell differentiation. Our data demonstrate that latent infection alters the differentiation state of CD34^+^ cells. Interestingly, while cGAS, STING, TBK1, and IRF3 protein levels were elevated in latently infected cells, the nuclear translocation of p-IRF3 was impaired, leading to reduced type I IFN expression. Notably, treatment with exogenous IFN-β restored myeloid differentiation, suggesting that latent HCMV suppresses differentiation by blocking p-IRF3 nuclear translocation and the subsequent type I IFN responses. These findings reveal a novel mechanism through which HCMV latency modulates host innate signalling and progenitor cell fate.

## 2. Materials and Methods

### 2.1. Viruses and Cells

HCMV strain RV-TB40-BAC_KL7_-SE-EGFP was kindly provided by Dr. K. Sampaio (University of Ulm) [68], and was propagated in the MRC-5 human lung fibroblast (cat. no. CCL-171; ATCC, Manassas, VA, USA) and Hs27 human foreskin fibroblast (cat. no. CRL-1634; ATCC) cell lines. The viral concentration of HCMV was determined as the number of plaque-forming units (pfu)/mL using a plaque assay and MRC-5 cells at 70–80% confluency. MRC-5 and Hs27 cells were maintained in Dulbecco’s Modified Eagle Medium (DMEM; cat. no. 11965084; Gibco, Waltham, MA, USA) with 10% foetal bovine serum (FBS; cat. no. 10270106; Gibco).

### 2.2. Peripheral Blood-Derived CD34^+^ Cell Culture Model

Peripheral blood mononuclear cells (PBMCs) were isolated from fresh whole blood samples from anonymous healthy human blood donors (Hong Kong Red Cross; ethics approval #REC/21-22/0217) using Lymphoprep (cat. No. 07851; STEMCELL Technologies, Vancouver, BC, Canada). CD34^+^ cells were then isolated from the PBMCs using a CD34 MicroBead Kit (UltraPure, human; cat. No. 130-100-453; Miltenyi Biotec, Bergisch Gladbach, Germany). The isolated CD34^+^ cells were cultured at a density of 2 × 10^4^ cells/200 μL in 96-well culture plates in Iscove’s Modified Dulbecco’s Medium (IMDM; cat. No. 12440046; Gibco) supplemented with 10% BIT9500 Serum Substitute (cat. No. 09500; STEMCELL Technologies), 50 μM 2-mercaptoethanol (cat. No. M6250; Sigma-Aldrich, St. Louis, MO, USA), 20 ng/mL human low-density lipoprotein (cat. No. SAE0053; Sigma-Aldrich), 1% penicillin–streptomycin (cat. No. 15140122; Gibco), 25 ng/mL granulocyte–macrophage colony-stimulating factor (GM-CSF; cat. No. 300-03; PeproTech, Cranbury, NJ, USA), 50 ng/mL interleukin 3 (IL-3; cat. No. 200-03; PeproTech), and 100 ng/mL stem cell factor (SCF; cat. No. 300-07; PeproTech) at 37 °C with 5% CO_2_, as described previously [69]. Every 3 to 4 days, 50% of the media was changed. The cells used for viral infection were cultured for at least 10 days.

### 2.3. HCMV Infection and Reactivation

Cells were spin-infected with HCMV diluted in DMEM at a multiplicity of infection (MOI) of 3 [15], centrifuged at 520× *g* for 45 min, and subsequently incubated at 37 °C with 5% CO_2_ for a total of 3 h. Infected CD34^+^ cells were cultured at a density of 1 × 10^5^ cells/200 μL in 96-well culture plates in IMDM complete medium without cytokines; 50% of the media was changed every 2 to 3 days. For reactivation, the cells were treated with 20 nM phorbol 12-myristate 13-acetate (PMA; cat. No. P8139; Sigma-Aldrich) on day 7 post-infection for 2 days, followed by co-culture with Hs27 monolayers for another 5 days to monitor for GFP fluorescent signals.

### 2.4. Stimulation of CD34^+^ Cells

For IFN-β stimulation, the cells were treated with 1000 U/mL or 3000 U/mL human IFN-β (cat. No. 300-02BC; PeproTech) for 3 and 7 days in IMDM complete medium without cytokines. For 2′3′-cGAMP stimulation, the cells were treated with 100 μg/mL 2′3′-cGAMP (cat. No. tlrl-nacga23; InvivoGen, San Diego, CA, USA) in IMDM complete medium without cytokines. The cells or supernatants were collected after 4, 6, and 20 h for RNA extraction, immunofluorescence confocal microscopy, and enzyme-linked immunosorbent assays (ELISAs), respectively. For brefeldin A (BFA) simulation, the cells were treated with 10 μM BFA (cat. No. inh-bfa; InvivoGen) in IMDM complete medium without cytokines. The cells were collected on day 1 for Western blot.

### 2.5. Live-Cell Imaging

The cells were observed using an Incucyte S3 Live-Cell Analysis system (Sartorius, Göttingen, Germany). Phase and fluorescence images were taken every 3 h. The green object count and total green object integrated intensity (GCU × μm^2^) of the infected cells were analysed using the Incucyte software v.2021B.

### 2.6. Flow Cytometry and Fluorescence-Activated Cell Sorting

The CD34^+^ cells were immunostained in phosphate-buffered saline (PBS; cat. No. 10010023; Gibco) with 1% FBS containing the following antibodies: anti-human CD34-VioBlue (cat. No. 130-113-74; Miltenyi Biotec), anti-human CD38-PE (cat. No. 555460; BD Biosciences, Franklin Lakes, NJ, USA), anti-human CD45RA-APC (cat. No. 130-113-362; Miltenyi Biotec), anti-human CD123-APC-Vio 770 (cat. No. 130-115-360; Miltenyi Biotec), and anti-human CD90-PE-Vio 770 (cat. No. 130-114-904; Miltenyi Biotec). Flow cytometry was performed using a BD FACSCanto II (BD Biosciences). For cell sorting, the cells were resuspended in PBS with 5% FBS and separated by gating based on GFP^+^ and GFP^−^ signals using the BD FACSMelody instrument (BD Biosciences). Analysis of the data was performed using FlowJo v.10.8.1 (FlowJo, LLC, Ashland, OR, USA).

### 2.7. Viral DNA, RNA Extraction, and Real-Time Quantitative PCR (qPCR)

DNA was extracted using NucleoSpin tissue XS (cat. No. 740901.50; Macherey-Nagel, Düren, Germany) according to the manufacturer’s instructions. Total RNA was extracted using RNAiso plus (cat. No. 9109; Takara, Tokyo, Japan) and then reverse transcribed to cDNA using a PrimeScript RT reagent kit with gDNA eraser (cat. No. RR047A; Takara) according to the manufacturer’s instructions. Gene expression was analysed with RT-qPCR using TB Green premix Ex Taq II (cat. No. RR820A; Takara) and a StepOnePlus real-time PCR system (Life Technologies, Carlsbad, CA, USA). The following primer pairs were used: *gapdh* (5′-ACAGTCCATGCCATCACTGCC-3′, 5′-GCCTGCTTCACCACCTTCTTG-3′); *ul55* (5′-GTAGCTGGCATTGCGATTGGT-3′, 5′-TCCAACACCCACAGTACCCGT-3′); *ul111.5a* (5′-GACTGGTTGTTGAGGCGGTA-3′, 5′-GCCGCCACAGAGACACTT-3′); *ul138* (5′-GGTTCATCGTCTTCGTCGTC-3′, 5′-CACGGGTTTCAACAGATCG-3′); *ul122* (5′-ATGGTTTTGCAAGCTTTGATG-3′, 5′-ACCTGCCCTTCACGATTCC-3′); *ifn-α* 1 (5′-GTCCTCCATGAGCTGATCCA-3′, 5′-GTTTCTCCCACCCTCTCCTC-3′); *ifn*-α 2 (5′-TGAATGACCTGGAAGCCTGT-3′, 5′-TTCTGCTCTGACAACCTCCC-3′); *ifn*-β 1 (5′-ACCAACAAGTGTCTCCTCCA-3′, 5′-GCAGTATTCAAGCCTCCCAT-3′); *cgas* (5′-TAACCCTGGCTTTGGAATCAAAA-3′, 5′-TGGGTACAAGGTAAAATGGCTTT-3′); *sting* (5′-CGGGTTTACAGCAACAGCAT-3′, 5′-GTGTCCGGCAGAAGAGTTTG-3′); and *ifi16* (5′-TAGAAGTGCCAGCGTAACTCC-3′, 5′-TGATTGTGGTCAGTCGTCCAT-3′). The viral DNA copy number was determined using a standard curve from dilutions of pCAGGS-UL55-myc containing the HCMV *ul55* sequence. Relative expression was calculated by normalising to *gapdh* expression using the ΔΔCT method. Further normalisation of the data from the control was performed accordingly.

### 2.8. Western Blot

Cells were washed with PBS and lysed with lysis buffer (10 mM Tris-HCl [pH 7.5], 200 mM NaCl, 1 mM EDTA, 1 mM DTT, 0.5% NP-40, 1 mM PMSF, 10 μg/mL leupeptin, 10 μg/mL aprotinin, 1.25 μg/mL pepstatin A, and 1× Halt phosphatase inhibitor cocktail [cat. No. 78427; Thermo Scientific, Waltham, MA, USA). For isolation of the cytoplasm and nuclear fractions, NE-PER nuclear and cytoplasmic extraction reagents (cat. No. 78833; Thermo Scientific) were used for cell lysis according to the manufacturer’s instructions. The protein concentration was then determined using a Pierce BCA protein assay kit (cat. No. 23225; Thermo Scientific). The protein samples were boiled in 5× loading buffer (250 mM Tris–HCl [pH 6.8], 30% glycerol, 10% SDS, 5% β-mercaptoethanol, and 0.02% bromophenol blue), separated by 10% SDS-PAGE, and transferred to Immobilon-P polyvinylidene difluoride (PVDF) membranes (cat. No. P2938; Merck Millipore, Burlington, VT, USA) using a mini trans-blot electrophoretic transfer cell (Bio-Rad, Hercules, CA, USA). The membranes were blocked with 5% blotting-grade blocker (cat. No. 1706404; Bio-Rad) and 0.5% bovine serum albumin (BSA; cat. No. A2153; Sigma-Aldrich) in TBST (20 mM Tris-HCl [pH 7.6], 150 mM NaCl, and 1% Tween-20) for 1 h at room temperature. They were then incubated with primary antibody diluted in 5% BSA in TBST overnight at 4 °C, and then incubated in the appropriate HRP-conjugated secondary antibody in TBST for 1 h at room temperature. The membrane blots were developed using the SuperSignal West Pico PLUS chemiluminescent substrate (cat. No. 34580; Thermo Scientific) or SuperSignal West Femto maximum sensitivity substrate (cat. No. 34094, Thermo Scientific). Band signals were detected using a ChemiDoc (Bio-Rad), and the intensities were quantified using Fiji software (ImageJ v.2.16.0/1.54p) [70]. The following primary and secondary antibodies were used: rabbit anti-cGAS (1:1000; cat. No. ABF124; Merck Millipore), rabbit anti-STING (1:1000; cat. No. ab181125; abcam, Cambridge, UK), rabbit anti-TBK1 (1:1000; cat. No. 38066; Cell Signaling Technology, Danvers, MA, USA), rabbit anti-phospho-TBK1 (Ser172) (1:1000; cat. No. 5483; Cell Signaling Technology), rabbit anti-IRF3 (1:1000; cat. No. ab68481; abcam), rabbit anti-phospho-IRF3 (Ser396) (1:1000; cat. No. 4947; Cell Signaling Technology), rabbit anti-lamin B1 (1:5000; cat. No. ab16048; abcam), rabbit anti-p44/42 MAPK (Erk1/2) (1:1000; cat. No. 9102; Cell Signaling Technology), rabbit anti-beta-actin (1:1000; cat. No. MABT523; Merck Millipore), HRP-conjugated donkey anti-rabbit antibody (1:10,000; cat. No. AP182P; Merck Millipore), and HRP-conjugated goat anti-mouse antibody (1:10,000 cat. No. AP130P, Merck Millipore).

### 2.9. RNAscope In Situ Hybridisation (ISH)

A total of 2 × 10^5^ cells/200 μL were seeded onto Nunc Lab-Tek II 8-well chambered coverglasses (Cat. No. 155409; Thermo Scientific) on day 4 post-infection. The cells were washed with PBS and fixed with 4% paraformaldehyde (cat. No. 158127; Sigma-Aldrich) for 15 min. HCMV RNA in situ hybridisation (ISH) was performed on the slides using an RNAscope Multiplex Fluorescent Reagent kit V2 (Cat. No. 323100; ACD, Newark, NJ, USA) following the manufacturer’s instructions. Briefly, the slides were hybridised with an HCMV-specific RNA probe for 2 h at 40 °C, followed by incubation with a cascade of signal amplification molecules (AMP1, AMP2, and AMP3) to amplify the signal and react upon binding of an opalTM fluorophore that targets the probe. The slides were then stained and mounted in ProLong Diamond Antifade Mountant with DAPI (cat. No. P36971; Invitrogen, Waltham, MA, USA). Confocal images were taken using a Leica TCS SP5 confocal microscope (Leica, Wetzlar, Germany) and analysed using LAS AF software v.3.0 (Leica). The percentage of the HCMV RNA+ cells was calculated as the number of RNAscope+ cells/total number of HCMV GFP-negative cells based on DAPI staining × 100%.

### 2.10. Immunofluorescence and Confocal Microscopy

A total of 2 × 10^5^ cells/200 μL were seeded onto Nunc Lab-Tek II 8-well chambered coverglasses (Cat. No. 155409, Thermo Scientific) on day 7 post-infection. The cells were washed with PBS, fixed with 4% paraformaldehyde (cat. No. 158127; Sigma-Aldrich) for 15 min, and permeabilised with 0.3% Tween-20 in PBS. The cells were blocked with 5% normal goat serum (cat. No. ab138478; abcam) and 3% BSA in PBST (0.01% Triton X-100 in PBS) for 1 h at room temperature, incubated with rabbit anti-phospho-IRF3 (cat. No. 37829; Cell Signaling Technology) diluted in 1% normal goat serum in PBST overnight at 4 °C, and then incubated with goat anti-rabbit IgG (H + L) highly cross-absorbed secondary antibody conjugated with Alexa Fluor Plus 647 (cat. No. A32733; Invitrogen) diluted in 1% normal goat serum in PBST for 1 h at room temperature. The cells were then stained and mounted in ProLong Diamond Antifade Mountant with DAPI (cat. No. P36971; Invitrogen). Confocal images were taken using a Leica TCS SP5 confocal microscope (Leica) and analysed using LAS AF software (Leica). The fluorescence signal intensity was measured using Fiji software [70].

### 2.11. Enzyme-Linked Immunosorbent Assay (ELISA)

The cell supernatants were collected on day 7 post-infection, and the concentrations of human IFN-β were measured using Human IFN-beta DuoSet ELISAs (cat. No. DY814-05; R&D Systems, Minneapolis, MN, USA) according to the manufacturer’s instructions. The absorbance was measured using a SpectraMax absorbance microplate reader (Molecular Devices, Silicon Valley, CA, USA).

### 2.12. Data Presentation and Statistical Analysis

The data were analysed using Prism 9 (GraphPad Software, Inc.). All graphs show the mean ± standard error of the mean (SEM) from at least three independent experiments. Statistical significance was determined using the two-tailed Student’s *t* test for the normalised experiments. Statistical significance was indicated with * *p* < 0.05; ** *p* < 0.01; and *** *p* < 0.001.

## 3. Results

### 3.1. Establishment of Latency in Primary CD34^+^ Cells Using RV-TB40-BAC_KL7_-SE-EGFP Virus

The RV-TB40-BAC_KL7_-SE-EGFP virus was used to infect peripheral blood-derived CD34^+^ cells. This virus produces GFP signals during productive infection driven by an attenuated MIEP, and a stronger signal intensity indicating cytopathic effects (CPEs) (Appendix A). When this virus was used to infect primary expanded CD34^+^ cell cultures at an MOI of 3, GFP^+^ cells (~50 in 1.6 × 10^4^) were detectable by day 1 post-infection (P.I.) using live-cell imaging (Figure 1A). The GFP^+^ cells appeared to be larger in size, while the morphology of the GFP^−^ cells was similar to that of the mock-infected cells (Figure 1A). The number of GFP^+^ cells and the intensity of the GFP signals reached a plateau between day 3 and 4 P.I. and at day 6 P.I., respectively (Figure 1B). These data indicate that HCMV replication occurred in the GFP^+^ cells, but no new GFP^+^ cells were observed at this time point. Using flow cytometry, the proportion of GFP^+^ cells was found to be ~0.5% of the total infected cell culture over time (Figure 1C). Based on the GFP signals expressed during productive infection, the GFP^+^ cells would likely be productively infected, and the GFP^−^ cells in the infected culture would likely harbour latent HCMV.

To determine if the GFP^+^ and GFP^−^ cells were productively and latently infected cells, respectively, we performed several characterisation experiments. First, we confirmed the presence of HCMV RNA in the GFP^−^ cells using RNAscope and a fluorescent probe targeting the *UL123* gene. Approximately 56% of the GFP^−^ cells showed positive signals when examined using confocal microscopy (Figure 1D). Second, using qPCR, the HCMV viral DNA copy number per cell was found to be around 150 and 2 for the FAC-sorted GFP^+^ and GFP^−^ cells, respectively (Figure 1E). Third, the GFP^−^ cells had relatively low levels of HCMV lytic gene expression compared with the GFP^+^ cells, with 74-, 354-, and 561-fold differences for the *ul122*, *ul55*, and *ul99* genes, respectively (Appendix A). Fourth, expression of the latent HCMV genes *ul111.5a* and *ul138* was detected in the GFP^−^ subpopulation, while expression of the productive gene *ul122* appeared to decrease between day 4 and 7 P.I. (Figure 1F). Fifth, to confirm the establishment of latency in the GFP^−^ cells, the cells were reactivated with 20 nM PMA for 2 days, followed by co-culture with Hs27 fibroblasts. GFP^+^ fibroblasts were found by day 5, indicating that reactivated GFP^−^ cells initiated productive infection and transmitted viruses to the fibroblasts, resulting in GFP^+^ CPEs (Figure 1G). In comparison, the co-culture of GFP^+^ cells (with no PMA pretreatment) with fibroblasts led to robust viral spread, indicating that these CD34^+^ cells were productively infected (Figure 1G). Therefore, the GFP^+^ cells (hereafter referred to as HCMV^GFP+^) following RV-TB40-BAC_KL7_-SE-EGFP infection represent productively infected cells, while the GFP^−^ cells (hereafter referred to as HCMV^GFP−^) are enriched for ‘truly’ latently infected CD34^+^ cells.

### 3.2. Differentiation State of CD34^+^ Cells Infected with RV-TB40-BAC_KL7_-SE-EGFP

To understand the cell differentiation states between productively infected HCMV^GFP+^ CD34^+^ cells and latently infected HCMV^GFP−^ CD34^+^ cells, we performed flow cytometry for the markers CD34, CD38, CD45RA, CD123, and CD90, as previously described [69]. The CD34^+^ HSPC subsets were examined using haematopoietic stem cell (HSC), multipotent progenitor (MPP), lymphoid-primed multipotent progenitor (LMPP), common myeloid progenitor (CMP), megakaryocyte–erythrocyte progenitor (MEP), granulocyte–monocyte progenitor (GMP), and common dendritic progenitor (CDP) markers (Figure 2A). The gating strategy for the analysis of the different cell subsets is depicted in Figure 2B. Over a seven-day time course, approximately 90% of the HCMV^GFP+^ cells were CD34^+^CD38^+^, which is in stark contrast to the mock-infected and HCMV^GFP−^ cells. Initially, approximately 50% of the mock-infected and HCMV^GFP−^ cells were CD34^+^CD38^+^, but by day 4 P.I., these percentages decreased to ~40% and ~30%, respectively (Figure 2C). Moreover, in the mock-infected cells, the major subsets observed on day 7 P.I. were MPPs, followed by CMPs, GMPs, and LMPPs, which were similar to those of the HCMV^GFP−^ cells (Figure 2D). Within the infected culture, the HCMV^GFP+^ cells were primarily myeloid progenitors, including CDPs (6.66%), GMPs (1.57%), and CMPs (1.28%) (Appendix A).

To further examine how latent infection affects the dynamics of the CD34^+^ subsets, we analysed their frequencies over time in the HCMV^GFP−^ cells. Except for HSCs on day 1 P.I., the frequencies of the CDP, CMP, and GMP cell subsets in the HCMV^GFP−^-infected culture were similar to those of the mock-infected cells except at the day 5 and day 7 timepoints (Figure 2D). The frequencies of the HSCs and LMPPs remained similar to those of the mock-infected cells on days 1 and 2 P.I. but showed increased proportions from day 3 P.I. onwards. MPPs initially decreased from day 1 to 3 P.I. but increased from day 4 onwards (Figure 2D). These data indicate that HCMV modulates the differentiation status of latently infected CD34^+^ cells. In addition, we observed marked differences in the differentiation state between the HCMV^GFP+^ and HCMV^GFP−^ CD34^+^ cells. This suggests that either productive infection occurs preferentially in more differentiated cells or viral replication drives cells towards a more mature phenotype. The removal of the productively infected GFP^+^ cells allowed for a more accurate experimental analysis of latent infection in CD34^+^ cells in the subsequent experiments.

### 3.3. Latent HCMV Impairs the Type I IFN-Induced Differentiation of CD34^+^ Cells

Type I interferons (IFNs) are known to induce differentiation of CD34^+^ cells during infection [48,49,71], but whether this process is modulated during latent HCMV infection remains unclear. Thus, we first examined the effect of type I IFNs (i.e., IFN-β) on CD34^+^ cell differentiation. Using flow cytometry, treatment with IFN-β at 1000 U/mL or 3000 U/mL was found to increase the frequency of CD38^+^ cells among CD34^+^ cells, with the myeloid progenitor CDP and CMP subsets increasing starting on day 3 post-treatment (Appendix A). In contrast, the frequency of the early progenitor HSC and MPP subsets decreased (Appendix A). Additionally, stimulation of the cells with 2′3′-cGAMP, which can activate the STING pathway and downstream IFN-β production (Appendix A), modestly increased the CMP subset by day 7 post-treatment and the GMP subset by day 3 post-treatment (Appendix A). Hence, our findings indicate that IFN-β treatment can promote myelopoiesis in our CD34^+^ cell model.

Next, we investigated whether the lack of differentiation in the HCMV^GFP−^ cells is associated with viral regulation of type I IFNs during latency. To assess this, we measured the mRNA expression levels of *ifna1*, *ifna2*, and *ifnb1*, as well as secreted IFN-β protein levels, in the mock-infected and HCMV^GFP−^ cells. As shown in Figure 3A, the HCMV^GFP−^ cells exhibited significantly decreased gene expression of *ifna1*, *ifna2*, and *ifnb* at day 3 P.I. This was further supported by decreased secreted IFN-β protein levels from the HCMV^GFP−^ cells at day 7 P.I., as measured using ELISA (Figure 3B). To investigate whether the addition of exogenous IFN-β could induce differentiation in latently infected cells, we treated the infected cultures with IFN-β. Interestingly, treatment with IFN-β increased the proportion of CD38^+^ cells within the HCMV^GFP−^ population compared with the untreated controls (Figure 3C). This suggests that the IFN-β treatment triggered myelopoiesis in the HCMV^GFP−^ cells, as evidenced by the increased proportion of myeloid subsets such as CDPs and GMPs (Figure 3D). These findings indicate that type I IFNs play a role in the differentiation of CD34^+^ cells towards the myeloid lineage, and the decreased proportions of CD38^+^ cells, CDPs, and GMPs in the HCMV^GFP−^ population can be partially rescued by the addition of IFN-β (Figure 2 and Figure 3). It is likely that latent HCMV inhibits this process to prevent the generation of immune-capable cells. However, the application of IFN-β appears to override this latent HCMV-mediated IFN-β inhibition, restoring cellular differentiation and allowing for future HCMV reactivation.

### 3.4. The STING Pathway Is Activated in Latently HCMV-Infected CD34^+^ Cells

To investigate how latent HCMV inhibits the expression of type I interferons (IFNs), we examined the activation status of the STING signalling pathway in CD34^+^ cells. STING, as a crucial adaptor protein in the innate immune response, is responsible for recognising viral DNA through DNA sensors and subsequently initiating the type I IFN response by activating TBK1, which in turn leads to IRF3 phosphorylation [29,72]. To our surprise, we observed elevated gene expression of *cgas* and *sting*, two key components of the STING pathway, in the HCMV^GFP−^ cells compared with the mock-infected cells (Figure 4A). We further confirmed this finding through Western blot analysis, which showed increased protein levels for cGAS and STING, as well as the downstream signalling pathway proteins TBK1 and IRF3, along with their phosphorylated forms in the HCMV^GFP−^ cells compared with the mock-infected control (Figure 4B).

One possible explanation for this increased expression could be differences in the baseline expression of STING pathway components among the various CD34^+^ cell subsets. To eliminate this possibility, we isolated uninfected CD34^+^ cells into two subsets, CD38^+^CD45RA^−^ ‘myeloid progenitors’ and CD38^−^CD45RA^−^ ‘early progenitors’, and conducted protein expression analysis. The Western blot analysis of cells from two donors showed that cGAS protein levels were slightly higher in the early progenitors than in the myeloid progenitors, although this difference was minimal in one of the two donor cells. Other STING pathway proteins showed similar expression between the two subsets (Appendix A). These data suggest that the observed increase in STING pathway proteins following latent HCMV infection is primarily infection induced rather than a reflection of intrinsic differences between cell subsets (Figure 4B).

To further confirm the activation of STING in HCMV^GFP−^ CD34^+^ cells, we conducted an experiment to evaluate the expression of p-TBK1 and p-IRF3 when STING translocation was inhibited by BFA. Compared with the untreated cells, we observed that the BFA treatment led to a decrease in p-TBK1 levels in both the mock-infected and HCMV^GFP−^ cells, with a more significant decrease observed in the latter. The levels of p-IRF3 remained similar to those of the mock-infected cells, but the increased levels seen in the HCMV^GFP−^ cells were restored to the mock levels upon STING inhibition (Appendix A).

Interestingly, despite clear activation of the STING pathway, we did not detect a corresponding increase in the production of type I interferons (IFNs) in the HCMV^GFP−^ cells (Figure 3A,B). This suggests that the activation of the STING pathway during latent HCMV infection does not have an effect on the production of type I IFNs. Since the STING/p-TBK1 pathway normally activates p-IRF3, leading to its translocation into the nucleus and subsequent transcription of type I IFN genes, it is possible that latent HCMV infection modulates this process.

### 3.5. Translocation of p-IRF3 Is Inhibited by Latent HCMV to Suppress Type I IFN Expression

To investigate the impact of latent HCMV on p-IRF3 translocation, we performed a confocal microscopy immunofluorescence analysis on the mock-infected and HCMV^GFP−^cells, with or without stimulation using 2′3′-cGAMP. In the absence of stimulation (control), the mock cells did not exhibit detectable p-IRF3 signals, while the HCMV^GFP−^ cells showed higher p-IRF3 signals in the cytoplasm compared to the mock-infected cells, but without a clear signal in the nucleus (Figure 5A). This observation aligns with the Western blot results, indicating increased activation of p-TBK1 and p-IRF3 in the HCMV^GFP−^ cells (Figure 4B). In contrast, the HCMV^GFP+^ cells displayed elevated p-IRF3 signals in both the cytoplasm and nucleus (Figure 5A), suggesting active p-IRF3 translocation in productively infected CD34^+^ cells. Upon 2′3′-cGAMP stimulation, an increase in the p-IRF3 signal was observed in both the mock-infected and HCMV^GFP−^ cells. However, nuclear p-IRF3 was clearly observed in the mock-infected and HCMV^GFP+^ cells, while in the HCMV^GFP−^ cells, the majority of the p-IRF3 signal remained cytoplasmic, with limited nuclear localisation (Figure 5A).

Analysis of the p-IRF3 signal intensities from the confocal images (Figure 5B) showed that the HCMV^GFP−^ and HCMV^GFP+^ cells showed higher levels of p-IRF3 compared with the mock-infected cells before stimulation with 2′3′-cGAMP (Figure 5C). After 2′3′-cGAMP stimulation, there was an increase in the level of p-IRF3 in the mock-infected cells, but there was no increase in the overall p-IRF3 signal in the HCMV^GFP−^ and HCMV^GFP+^ cells compared to the untreated cells (Figure 5C). To determine if p-IRF3 translocation occurred, the nuclear-to-whole cell (N:C) ratio of the p-IRF3 raw intensities was analysed. The N:C ratio of the mock-infected and HCMV^GFP−^ cells was similar before stimulation, indicating that there is no increase in p-IRF3 translocation in the HCMV^GFP−^ cells (Figure 5D). Increased translocation was found after 2′3′-cGAMP stimulation in the mock-infected cells, but not in the HCMV^GFP−^ cells (Figure 5D). In contrast, the HCMV^GFP+^ cells had a higher N:C ratio in both the unstimulated and stimulated states compared with the mock-infected cells (Figure 5D). The Western blot analysis confirmed the results, with similar levels of cytoplasmic p-IRF3 between the mock-infected and HCMV^GFP−^ cells but a marked decrease in the nuclear level (Figure 5E). The levels of IRF3 and the cytoplasmic control ERK remained similar. However, both the p-IRF3 and IRF3 levels in the nucleus of the HCMV^GFP−^ cells appear to be lower than those of the mock-infected cells, while the levels of the nuclear lamin B1 control were similar (Figure 5E). Taken together, these findings demonstrate that latent HCMV infection activates the STING/p-TBK1/p-IRF3 signalling cascade but impairs p-IRF3 nuclear translocation. This restriction likely accounts for the diminished type I IFN expression and suppression of CD34^+^ cell differentiation.

## 4. Discussion

Despite the presence of an intact anti-HCMV immune memory, latent HCMV in CD34^+^ HSPCs exhibits various characteristics that ensure its survival within the human host. Consequently, the virus remains a lifelong threat as an opportunistic pathogen capable of causing life-threatening diseases in immunosuppressed or immunocompromised individuals [73]. The low frequency of HCMV-DNA^+^ CD34^+^ cells and the lower percentage of latent transcript expression enable the virus to effectively evade detection [74]. Latent HCMV-mediated immune evasion may be attributed to the impairment of CD34^+^ HSPC differentiation, thereby preventing the development of professional antigen-presenting cells like dendritic cells. Our study demonstrated one mechanism through which latent HCMV can influence the differentiation of CD34^+^ myeloid cells: inhibiting p-IRF3 nuclear translocation-mediated type I IFN responses downstream of the innate immune STING/p-TBK1 pathway.

STING serves as an adaptor protein for the innate immune type I IFN response and can be activated by various cytosolic sensors, including cGAS, DNA-dependent activator of IRFs (DAI), DDX41, and IFI16 [55,57,75]. Activation of the cGAS/STING pathway can impact downstream effector molecules such as IRFs, NF-κB, mitogen-activated protein kinases (MAPKs), and mammalian target of rapamycin (mTOR), which, in turn, affect the proliferation, mobilisation, or differentiation of HSCs [52]. Recognition of bacterial pathogen-associated molecular patterns (PAMPs) with STING leads to the phosphorylation of IRF3, subsequently reducing the number and repopulation capacity of long-term HSCs [53]. The cGAS/STING pathway can efficiently recognise HCMV DNA during productive infection of differentiated cells like fibroblasts and dendritic cells, triggering antiviral type I IFN responses [72,76]. However, it remains unknown whether the cGAS/STING pathway can recognise latent HCMV DNA and induce cell differentiation. Therefore, our data provide evidence that latent HCMV regulates the translocation of p-IRF3, inhibiting the STING/IFN-I pathway to prevent myelopoiesis. Interestingly, latent HCMV upregulates cGAS, STING, (p-)TBK1, and (p-)IRF3 proteins but selectively inhibits the crucial step of p-IRF3 translocation to avoid the production of type I IFNs. This contrasts with productive infection, where at least nine proteins, including US9, UL31, UL35, UL42, UL48, UL82 (pp71), UL83 (pp65), UL94, and UL122, are known to inhibit the STING pathway [76,77,78,79,80,81,82,83,84].

Among these proteins, only US9 has been shown to impede the translocation of p-IRF3 during productive infection [76]. Notably, US9 is not expressed during latent infection, and a recent study demonstrated that the latent protein UL138 can directly bind to both STING and TBK1, leading to the degradation of STING in lysosomes and the inhibition of TBK1 [29]. This likely results in reduced accumulation of p-IRF3 and a subsequent decrease in IFN-β expression, as demonstrated in productive infection or overexpression models. However, during latent infection, when viral protein expression is minimal, the effect of UL138 may be partial and likely requires the involvement of other latent proteins to act in concert. It is possible that the virus employs other latent proteins to strategically inhibit the final step of the STING pathway, preventing the nuclear translocation of p-IRF3 and the induction of type I IFNs. Therefore, the identification of the specific latent HCMV protein(s) responsible for preventing p-IRF3 translocation would be crucial for future studies.

The activation of the STING pathway upstream of p-IRF3 during latent infection in CD34^+^ cells suggests the activation of DNA sensors [85,86]. It is likely that the sensing of latent HCMV involves its episomal circular DNA, which is tethered to the host chromosomes [85,86]. DNA sensors that function in the nucleus, such as IFI16 and cGAS, have been shown to mediate interferon responses through STING, leading to p-IRF3 activation and translocation [87,88]. In our study, we observed an increase in both cGAS and IFI16 expression in latently HCMV-infected CD34^+^ cells (Figure 4B, Appendix A). Furthermore, recent research has demonstrated that STING is present in the nucleus, where it interacts with a group of DNA- and RNA-binding proteins, some of which may also bind to IRF3 and IRF7 [89]. Therefore, the presence of DNA sensors in the nucleus suggests that they may continuously recognise latent HCMV DNA and initiate downstream signalling. However, HCMV-controlled interferon responses hinder the progression of this signalling cascade by preventing p-IRF3 translation, thereby impeding the differentiation response of these cells. In other words, CD34^+^ myeloid progenitors are primed to become efficient antigen-presenting cells that stimulate the antiviral immune response, but latent HCMV exerts regulatory control over this process.

The fate of latently HCMV-infected cells and their potential to differentiate into antigen-presenting cells remains uncertain. Previous reports have indicated that latent HCMV infection in myeloid cells leads to an immunosuppressed profile characterised by high expression of B7-H4 or low expression of CD74 [22,44]. In our study, we demonstrated that HCMV may suppress type I interferon production as a means to limit CD34^+^ cell differentiation. This likely facilitates the establishment of a latent infection by preserving progenitor cell viability and preventing premature activation, thereby maintaining a reservoir for future reactivation. Our findings also suggest that this control over the STING/p-IRF3 axis may be reversible under certain conditions, for example, by increasing the protein levels of related innate immune signalling pathway components or inhibiting viral proteins like UL138. The use of 2′3′-cGAMP alone was insufficient to achieve this, suggesting that an alternative approach may be necessary to counteract viral control of p-IRF3. Nevertheless, stimulating latently infected CD34^+^ cells with IFN-β resulted in increased frequencies of CDPs, CMPs, and GMPs (Figure 3). It would be crucial to evaluate the functions of these differentiated cells in terms of phenotypic and genetic changes, as well as their capacity for antigen presentation to activate HCMV-specific T cells.

By utilising the RV-TB40-BAC_KL7_-SE-EGFP virus, we can distinguish between CD34^+^ cells that support productive infection and those that support latent infection. It is important to note that our model induced latency in approximately 56% of the HCMV^GFP−^ population, as determined by the RNAscope analysis (Figure 1D). A key limitation of our model is that the HCMV^GFP−^ population likely consists of a heterogeneous mixture of truly latently infected cells, abortively infected cells, and uninfected bystanders, which could have caused the minimal differences in our experiments. Moreover, productive infection occurs in a small subset (<1%) of CD34^+^ cells, specifically the CDP, CMP, and GMP subsets. This raises concerns as the analysis of viral gene expression may include cells with different latency statuses and lytic gene expression patterns. Interestingly, a recent study showed that HCMV latency was found in CD38^+^ CD90^+^ and CD38^+/low^ CD90^−^ CD34^+^ HPCs, and lytic infection was observed in CD38^−^ CD90^+^ cells [61]. In this study, we found that lytic infection predominantly occurred in CD38^+^ cells. These differences may be due to the variations in the source of progenitor cells or the viral infection model used. In order to obtain more accurate data on latent HCMV gene expression profiles, it is crucial to compare single-cell RNA sequencing data from HCMV^GFP−^ and HCMV^GFP+^ CD34^+^ cells in extended culture. Incorporating the expression of the GFP gene using RV-TB40-BAC_KL7_-SE-EGFP will facilitate improved transcriptional profiling, allowing for better differentiation between productive and latent infections.

Finally, we observed unhindered p-IRF3 translocation in the HCMV^GFP+^ cells compared with the HCMV^GFP−^ cells, suggesting that during productive infection, the host cells can overcome the HCMV control of p-IRF3 and viral immune evasion. Interestingly, we observed a slight reduction in the total p-IRF3 signal and diminished nuclear translocation in the HCMV^GFP+^ CD34^+^ cells following 2′3′-cGAMP stimulation. This may reflect either the feedback suppression of the STING/IRF3 axis or enhanced viral antagonism. This is supported by a recent study that demonstrated that, although HCMV initially activates the STING pathway to promote immediate-early gene expression, it later suppresses downstream IFN-β and ISG responses via multiple viral antagonists [61]. Thus, HCMV may transiently benefit from early innate signalling but quickly shuts it down to maintain control.

In conclusion, during latency in CD34^+^ cells, HCMV inhibits the STING pathway by regulating downstream p-IRF3 nuclear translocation, thereby preventing the expression of type I IFNs. This lack of a type I IFN response hinders the differentiation of these cells into competent antigen-presenting cells, ensuring the virus’s survival within these latent reservoirs. It is crucial to identify the specific HCMV gene(s) or protein(s) responsible for this inhibition in order to devise a strategy to overcome the control exerted on p-IRF3.

## Figures and Tables

**Figure 1 viruses-17-01109-f001:**
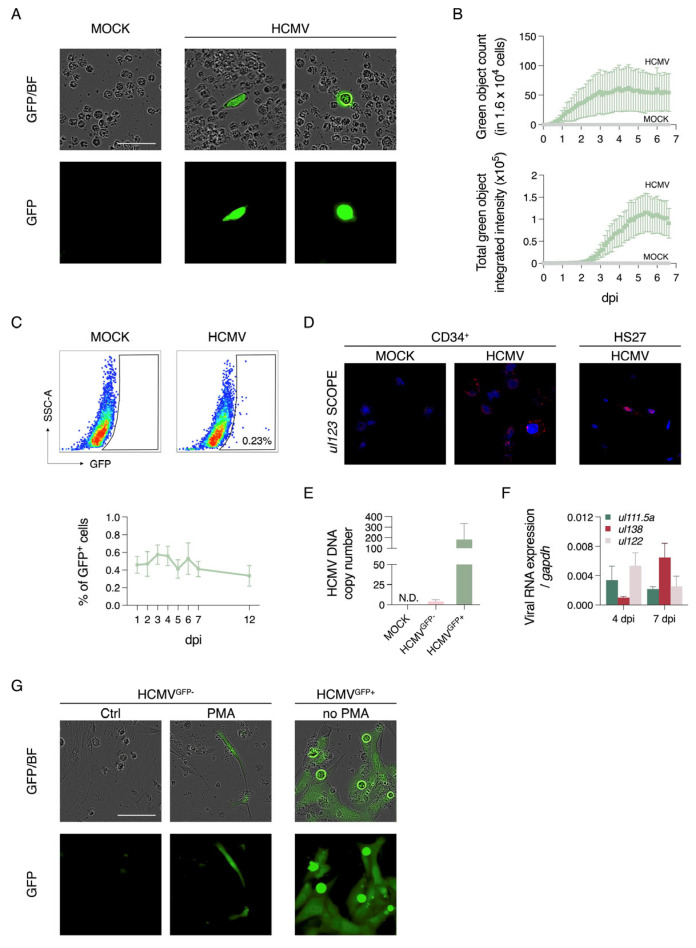
HCMV strain RV-TB40-BAC_KL7_-SE-EGFP-infected CD34^+^ cells. CD34^+^ cells were infected with HCMV (RV-TB40-BAC_KL7_-SE-EGFP) and monitored by live-cell imaging over 7 days. (**A**) Representative images of the mock- and HCMV-infected cells on day 4 P.I. Scale bar = 50 µm. (**B**) Quantification of the green object count and total green object integrated intensity over time (*n* = 6). (**C**) Percentage of GFP^+^ cells as measured by flow cytometry. Representative dots plot from day 7 P.I. are shown alongside a line graph of GFP^+^ cell percentages over time (*n* = 3). (**D**) RNAscope analysis targeting *UL123* in the HCMV-infected CD34^+^ cells and Hs27 fibroblasts. Red fluorescent puncta indicate positive hybridisation. (**E**) Quantification of HCMV DNA copy number per cell using qPCR targeting *UL55* in the FAC-sorted HCMV^GFP+^ and HCMV^GFP−^ subsets on day 7 P.I. (*n* = 3). N.D., not detected. (**F**) RT-qPCR analysis of relative expression of the viral genes *ul111.5a*, *ul138*, and *ul122*, normalised to *gapdh*, on days 4 and 7 P.I. (*n* = 3). (**G**) Reactivation of the HCMV^GFP−^ cells was induced using 20 nM PMA for 2 days, followed by co-culture with Hs27 fibroblasts for 5 days. HCMV^GFP+^ cells co-cultured without PMA treatment were used as a control. Graphs show the mean ± SEM from the indicated number of independent experiments.

**Figure 2 viruses-17-01109-f002:**
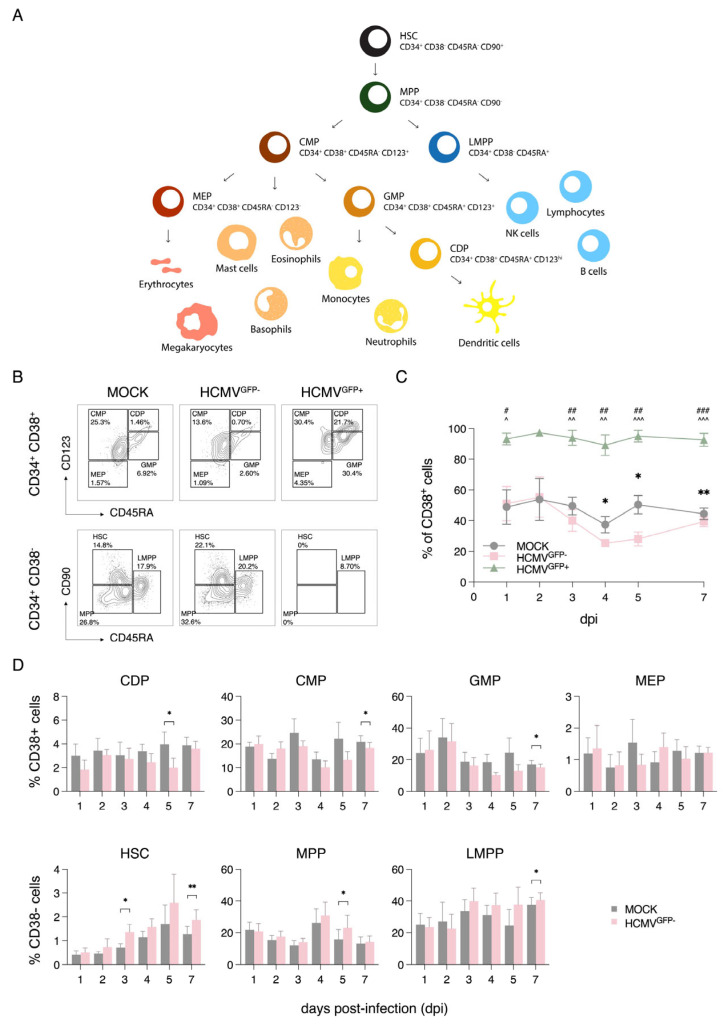
Differentiation state of CD34^+^ cells infected with RV−TB40−BAC_KL7_−SE−EGFP. (**A**) CD34^+^ haematopoiesis with indicated cell markers to identify the following subsets: haematopoietic stem cell (HSC), multipotent progenitor (MPP), lymphoid-primed multipotent progenitor (LMPP), common myeloid progenitor (CMP), megakaryocyte–erythrocyte progenitor (MEP), granulocyte–monocyte progenitor (GMP), and common dendritic progenitor (CDP) subsets. (**B**) Gating strategy for analysing the subsets in CD34^+^CD38^+^ and CD34^+^CD38^−^ cells. (**C**) Percentage of CD38^+^ cells and (**D**) percentage of CD34^+^ progenitor subsets in the mock- and HCMV-infected cells over time (1, 2, 3, 4 and 5 dpi: *n* = 3; 7 dpi: *n* = 7). At least 10,000 events were recorded per sample. Graphs show the mean ± SEM from the indicated number of independent experiments. Statistical tests were performed between mock vs. HCMV^GFP+^ (#), HCMV^GFP−^ vs. HCMV^GFP+^ (^), and mock vs. HCMV^GFP−^ (*). *^, #, ^^
*p* < 0.05, **^, ##, ^^^
*p* < 0.01, ^###, ^^^^
*p* < 0.001.

**Figure 3 viruses-17-01109-f003:**
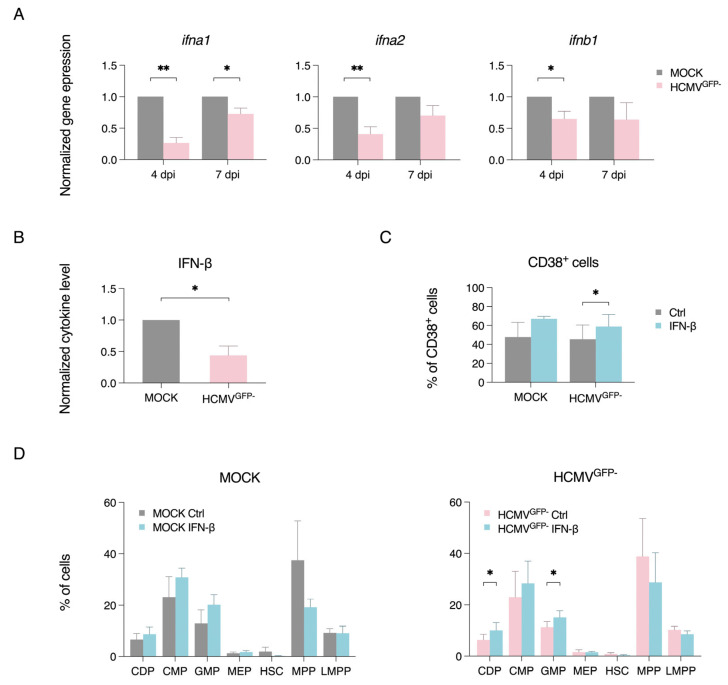
Type I IFNs are inhibited in HCMV^GFP−^ cells, altering the differentiation of CD34^+^ cells. Total RNA of the mock-infected and HCMV^GFP−^ cells was collected on day 7 P.I. (**A**) RT-qPCR analysis of relative expression of type I IFNs, *ifna1*, *ifna2*, and *ifnb1* normalised to the mock-infected cells (*n* = 3). *gapdh* served as the housekeeping gene. (**B**) Quantification of secreted IFN-β protein in cell supernatants collected on day 7 P.I., measured by ELISA and normalised to mock infection (*n* = 3). (**C**) Percentage of CD38^+^ cells in the mock-infected and HCMV^GFP−^ cells with or without IFN-β treatment (*n* = 4). (**D**) Percentage of CD34^+^ progenitor subsets in the mock-infected (left) and HCMV^GFP−^ cells (right) with or without IFN-β treatment. Graphs show the mean ± SEM from the indicated number of independent experiments. * *p* < 0.05, ** *p* < 0.01.

**Figure 4 viruses-17-01109-f004:**
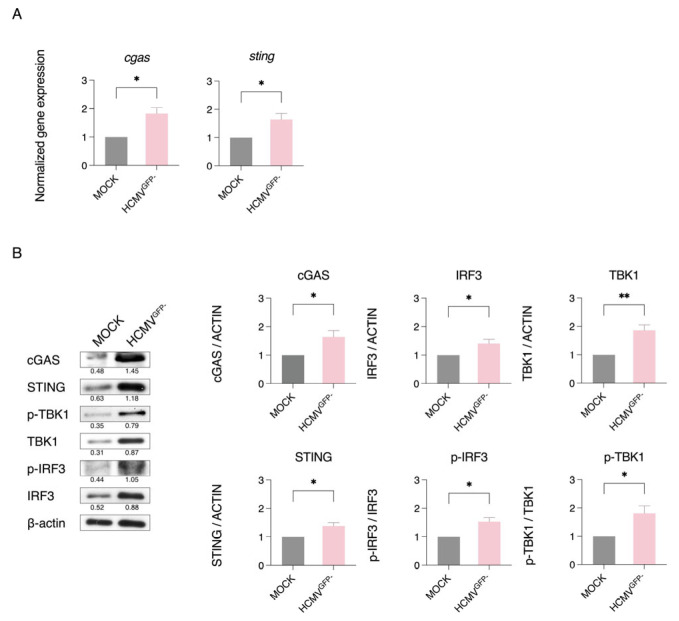
STING/p-TBK1/p-IRF3 pathway is activated in latently HCMV-infected CD34^+^ cells. (**A**) RT-qPCR analysis of the relative expression of *cgas* and *sting* in HCMV^GFP−^ cells normalised to the mock-infected cells (*n* = 3) at day 7 P.I. (**B**) Western blot analysis of cGAS, STING, p-TBK1, TBK1, p-IRF3, and IRF3 protein expression normalised to mock infection (*n* ≥ 3). β-actin served as the endogenous control. Numbers under the blot indicate the band intensities for the different proteins normalised to β-actin. The band intensities for the different proteins normalised to β-actin are shown from three independent experiments. Graphs show the mean ± SEM. * *p* < 0.05, ** *p* < 0.01.

**Figure 5 viruses-17-01109-f005:**
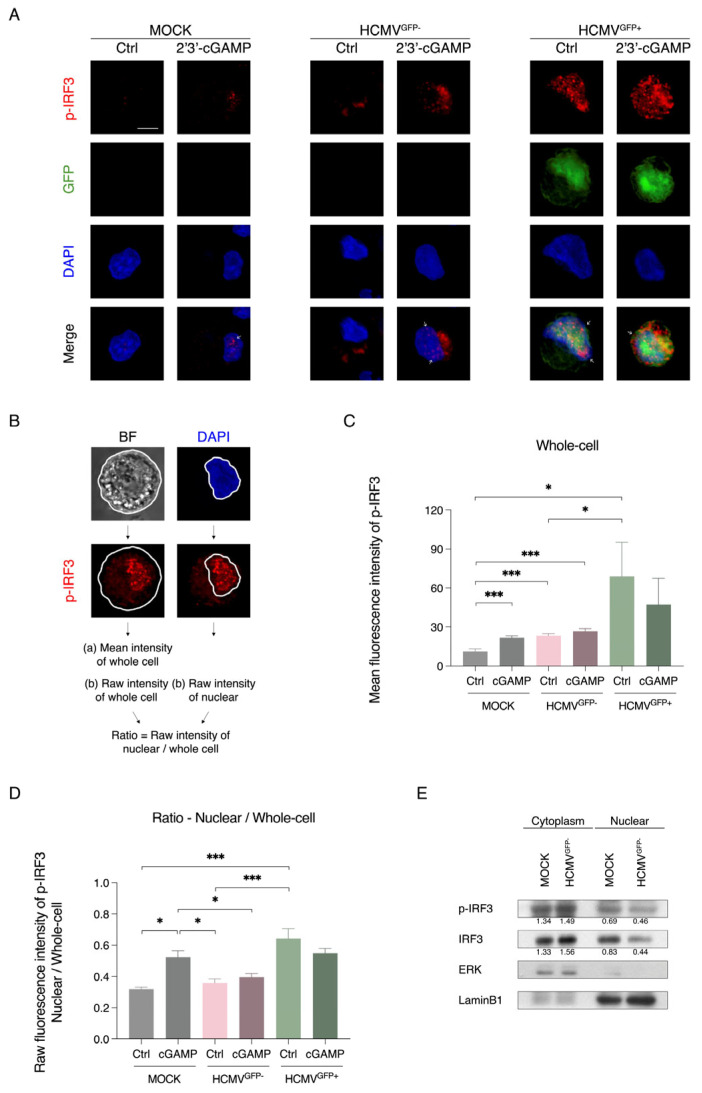
Nuclear translocation of p-IRF3 is inhibited by latent HCMV. Mock- and HCMV-infected cells were treated with 100 µg/mL 2′3′-cGAMP for 6 h to activate the STING pathway. Immunofluorescence was performed using specific antibodies against p-IRF3 (red), with DAPI (blue) used as a marker for the nucleus and GFP as a marker for HCMV^GFP+^ cells. (**A**) Representative images of unstimulated cells (Ctrl) and 2′3′-cGAMP-stimulated mock-infected, HCMV^GFP−^, and HCMV^GFP+^ cells. Scale bar = 5 µm. (**B**) p-IRF3 signals in the nucleus and cytoplasm measured based on the perimeters of the cell and nuclear membrane from brightfield and DAPI staining images. (**C**) Mean p-IRF3 fluorescence intensity for the whole cell for the different treatments. (**D**) Ratio of nucleus to whole cell raw p-IRF3 intensities indicating the degree of p-IRF3 translocation. (**E**) Western blot analysis of p-IRF3 and IRF3 protein expression in cytosolic and nuclear fractions. Lamin B1 served as the endogenous nuclear control; ERK served as the endogenous cytoplasm control. Numbers under the blot indicate the band intensities for the different proteins normalised to corresponding control. Graphs show the mean ± SEM from at least five biological replicate experiments. * *p* < 0.05, *** *p* < 0.001.

## Data Availability

The datasets analysed during the current study are available from the corresponding author upon reasonable request.

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
