# Peer review of "Latent Human Cytomegalovirus Infection Activates the STING Pathway but p-IRF3 Translocation Is Limited"

_viruses, 2025, doi:10.3390/v17081109_

Round 1
Reviewer 1 Report
Comments and Suggestions for Authors
The authors aim to investigate how HCMV modulates the differentiation state of latently infected CD34+ cells, which is an important topic that can provide insight into HCMV-induced pathogenesis. However, I have significant concerns regarding the generation of the HCMV latency model presented in this study.
Specifically, the authors differentiate productive and latent infections based on the presence or absence of GFP signals in primary infection. They report that only approximately 0.5% of the CD34+ cells were GFP-positive after HCMV infection, categorizing these cells as productively infected, while the GFP-negative cells were designated as latently infected (lines 281-283). This classification raises questions about the evaluation of infection efficiency in this assay. The remaining ~99.5% of CD34+ cells lacking GFP expression may not be infected at all, as the true infection efficiency cannot be assumed to be 100%.
Furthermore, in Fig 1G, the authors indicate that the observed GFP signals are driven by an attenuated MIEP (lines 271-272). Thus, the detection of GFP expression after PMA treatment cannot robustly confirm the reactivation of HCMV latency. It would be more appropriate to investigate viral lytic or structural antigens to validate reactivation and provide stronger evidence of the successful generation of HCMV latency.
Reviewer 2 Report
Comments and Suggestions for Authors
Understanding how HCMV establishes latency is key to understanding pathogenesis. How the virus controls CD34+ progenitor cells (specifically myeloid lineage differentiation) is a central and longstanding question to understanding the fundamentals of latency. Lee, Ye, and Cheung address this through a new study looking at phospho-IRF3 translocation specifically in myeloid lineage cells derived from HPCs.
Main points:
- The cited literature along with the introduction and discussion appears somewhat superficial, for example (and as examples only, there are multiple places):
- For example: newer work is not included (e.g., more recent review articles covering the scope of what is known about HCMV (most recent citation in this area is from 2021/2022, and most are from before 2016 – while multiple newer reviews from several different groups were published in the last couple of years, including updated reviews from some of these same authors)
- Information on the known role of IRF3 / interferon response, both within latency and during other infection stages is missing.
- Information on the known latency factors are not complete (e.g., missing information on US28 (could be replaced by recent review if necessary), most of the UL138 literature is not cited nor discussed, none of the UL135 and UL136 literature is included.
- Information on the role of miRNAs in latency is cited as one paper (there are multiple and the paper cited does not represent all the information the authors included in lines 70-71)
- This paper focuses on IRF3 – however, background/prior information on HCMV and IRFs is described minimally or is missing here (e.g., DeFillppis 2009, 2010, Ren 2023, Pham 2021, Ashley 2019, Hare 2020, Elder 2019 (primary paper) – as examples to start that have direct connections to IRF3 and/or latency).
- The role of IFN-beta (specifically blocking IFN-beta) as used here to alter latency (in viral and bacterial systems) and as a mechanism behind myelopoiesis has been described elsewhere and should be discussed, including that of Schnayder 2020 which the authors cite but do not discuss in this context.
While this is most likely an oversight, for the relatively broad audience of this journal, these oversights give a misleading impression of the literature and are needed to place this study into the appropriate context.
- Key papers discussing progenitor cell subsets and their role in HCMV latency are missing: Crawford et. al., 2021, Cheung et. al., 2017, Sindre et. al., 1996; and an additional paper (Goodrum et. al., 2022) is cited but not discussed. As the data (for example, analysis of the CD34 populations such as these authors repeat in their Figure 2, analysis of gene expression in different (GFP+ vs GFP-) subpopulations such as these authors Figure 1, etc) in these papers is directly applicable to the topic of this paper, this oversight is highly disconcerting.
- The viral strain and standard culture conditions are well described. However, it is misleading to describe the cells used in most of the paper as CD34+ HSCs when as the authors describe these are PBMC total CD34+ cells. This population does include HSCs (a minority population), as well as many cells of more mature lineages including in many cases cells such as endothelial progenitors (which are not hematopoietic). In lines 128-137, the authors also describe that their cells are culture for “at least 10 days” in cytokine containing media which includes GM-CSF used to differentiate HS/HPCs into monocyte lineages. I do not see data which confirms that these cells remain CD34+ without differentiation in this system nor that they retain stem cell function (e.g., colony forming ability and/or engraftment), and this cocktail, while previously described (e.g., Egeland et al 1991 as a first example) to maintain CD34+ numbers does not maintain function. This may also be illustrated by the relatively low level of infection in these CD34+ cells (line 274 “~50 in 1.6 x 10^4” or ~0.3% (listed “~0.5% in the total infected cell culture overtime” on lines 280-281 by flow) when prior published papers state on the order of 5-45% infection of CD34+ cells (Goodrum, Nelson, Sinclair, Stern-Ginossar, and other labs). While Figure 2 assess markers of stem cell identity, it is not clear if lineage+ cells were gated out or they are also included (e.g., cells that have matured beyond a stem cell (HSC, CMP, etc) fraction but still retain surface expression of early receptors – as is common during transition). As the authors clearly describe their results as occurring in CD34+ cells and HSCs throughout the paper, this discrepancy should be discussed and/or data demonstrating what these cells are should be included, or language (throughout including abstract and title) adjusted to clarify CD34+ but not HSCs as appropriate for the authors’ overall story.
- Latency establishement:
- What is the change in latent gene expression & latency phenotype between GFP+ and GFP- cells in Figure 1? The authors compare DNA copy number using two methods (great!) and the classic lytic gene expression (in supplemental). They also show latency-associated genes in Figure 1 for the GFP- cells, though this is not included for the GFP+ cells to determine if there is a difference in these populations (which would help validate the other information). It is also key to determining if the lower expression of lytic gene expression is due to latency establishment vs overall reduced infection in these cells (e.g., if there is a higher ratio of latent:lytic gene expression or if latent gene expression is equivalent in GFP- and GFP+ but lytic gene expression is lower in the GFP-).
- Also, and perhaps more importantly, it is not clear if latency was established in either population. The data in Figure 1G shows both PMA treated and untreated for the GFPnegative population, showing visually an apparent increase. However, given the numbers reported earlier in the figure, without quantification it is hard to tell if this is “all” the cells that were infected or select. Most importantly, the GFP+ fraction shows no apparent change in GFP expression with or without PMA treatment, suggesting that viral replication is ongoing (no latency) and no data on whether infectious virus is produced during the latency culture period (or if it increases following PMA treatment) is included.
- What effect on cgas/sting genes and the overall pathway was observed in GFP+ CD34+ cells? (Figure 4) Given that the cellular populations between Mock and GFP- fractions are distinct (as described by the authors in Figure 2), is the cellular identity altering the readout for cgas/sting or not? (e.g., Xu 2025, Smith 2016, Demerdash 2021, etc – role of IFNs in HSCs in the absence of infection where IFN both alters stem cell subset proportions and different stem cell subsets have altered cgas/sting signaling).
May be main or minor points depending on how the language/information is revised and if this (post-revision) changes the scientific meaning of the sentences:
- Grammatical clarity on lines: 18-19, 427, 452-454, plus other locations for general clarity but for which I do not believe would significantly alter the scientific meaning.
- for example what does this mean? “Over a seven-day time-course experiment, the proportion of latently infected CD34+ cell subsets of myeloid progenitors was mock infection”. How is “mock infection” a proportion of a whole?
- Line 427 “control as media”. Is media was used as a control (no? as Mock cells were listed as the control).
- Lines 452-454: “Interestingly, nuclear p-IRF3 was observed in mock and HCMVGFP+ cells, but not in HCMVGFP- cells, where most p-IRF3 loci were found outside the nucleus (Figure 5A).” This is supported by Figure 5A in single cell representative images. However, Figure 5D quantifies nuclear: whole cell (rather than nuclear:cytoplasmic which would support this data), and seems to imply (with ratios of ~0.5 nuclear: whole cell) that about half the protein is located in the nucleus in Mock cGAMP, slightly more in GFP+, but about 0.4 both ctrl and cGAMP GFP-. If ~40% of the protein is still nuclear in GFP- cells, how is most p-IRF3 found outside the nucleus? Figure 5E may also support a slight change away from nuclear p-IRF3 in GFP- compared to mock cells, however, no quantification or loading control (for total input) is included in this figure.
- Information on the number of cells analyzed in Figure 2 should be included.
- Supplementary western blot figures still appear to be cropped – full blots should be included here for best data practices (and I believe for this journal’s rules)
- An IRB statement as these are patient samples is required (I believe even if they have been reviewed to not need IRB, this needs to be included).
Minor comments / additional suggestions (note suggestions, by this review – I consider these interesting, but not essential to the paper)
- The authors discuss a potential change in morphology and/or size of infected cells (Figure 1, discussion lines 275-277), it would be an interesting addition of this size change could be quantified using their image analysis platform.
Overall language is fine with the exception of several confusing points described in the main review (as changes in language in those areas may alter scientific interpretation).
Reviewer 3 Report
Comments and Suggestions for Authors
Please see the attachment

The result and figure legend could use more details related to the experiments.
Reviewer 4 Report
Comments and Suggestions for Authors
This work from Wang Ka Lee and colleagues focuses on the impact of HCMV infection on the differentiation state of CD34+ myeloid progenitor cells. The relationship between HCMV and the haematopoietic system is well established, but difficulties studying these events remain in the field, due to the combination of the difficulty of working with CD34+ myeloid progenitor cells and the need to use refined approaches to avoid combining multiple HCMV lifecycles (latency, productive lytic, abortive lytic) in the analyses. The authors here utilise isolated CD34+ cells for all their analyses and a GFP-expressing HCMV strain to ensure that they can separate lytically infected cells from the rest of the population. Generally I find the manuscript to be well written, the data of interest and the conclusions supported by the data.
Figure 1 primarily focuses on the validation of the model system, demonstrating that GFP+ CD34+ cells readily produce infectious HCMV that can infect fibroblasts, that a majority of GFP- cells contain HCMV genomes, and that GFP- cells express an expected pattern of viral gene expression (i.e. expression of well-established latency-associated genes in the relative absence of IE gene expression). Crucially, the authors show that treatment of GFP- cells with PMA leads to the production of infectious HCMV, in line with effective reaction. The system is well characterised, but the authors should be cautious in asserting that ‘all’ of the GFP- cells represent truly latently infected cells. Without some form of single-cell analysis or showing that infectious virus can be recovered from a vast majority of these cells upon reactivation, it is possible that many of the GFP-, HCMV DNA+ cells represent ‘dead ends’ in which the viral genome is fully repressed and not able to manipulate the host cell nor reactivate. This might explain the relatively small differences often found in later figures. An additional interesting insight that the authors may wish to consider is whether the PBMCs of the donors (or specific subsets) are capable of activating in response to stimulation from the GFP- cells (this would require the donors to be HCMV positive individuals). A lack of immune response would highlight the fully ‘latent’ state of these cells, as experimentally induced latency is obviously distinct from in vivo latency and is likely to include cells that would have otherwise been deleted by the immune system. This could even function as a method to focus in on cells that are more likely to reflect in vivo latent cells, as immune cells could be used to deplete ‘noisy latent’ cells that would fail to persist in vivo.
Figure 2 focuses on the differentiation state of CD34+ cells in response to HCMV infection. The authors find that lytic infection is primarily limited to CD38+ cells, whilst the GFP- population much more closely resembles that of the mock population, albeit with a small decrease in the number of CD38+ cells in the GFP- population. The authors should double check their statistics, as the indicated ** for the difference between day 7 GFP- and mock is surprising given the lower significance observed at earlier times for greater differences, although this could be due to more consistent data, which is difficult to judge given that the error bars plotted are SEM as opposed to the more informative SD. The authors also state that GFP+ cells represent a more matured phenotype likely due to viral replication, but it is equally possible that the cells that became GFP+ were already more mature and that contributed to the establishment of a productive infection, as opposed to replication driving them down the pathway. It is also important to note that all these analyses are based on frequencies as opposed to total numbers, so that ‘increases’ in populations can be driven by loss of other populations, though I do not believe this overly impacts the conclusions drawn.
Figure 3 addresses the impact of HCMV infection on the ability of cells to differentiate in response to IFN mediated stimuli. They show reduced expression of IFNs in GFP- cells, and that IFNb treatment can lead to differentiation of both mock and GFP- cells to a similar extent. The authors should acknowledge that exogenous application of IFN seems to override HCMV-mediated repression of IFN expression, at least in relation to cellular differentiation status. This potentially aligns with the needs with the virus, as during seeding of the latent compartment the virus seeks to inhibit IFN production to reduce cellular differentiation and anti-viral activities, leading to a greater number of successful latent infections, but ultimately the virus must retain the ability to reactivate, so that when a different inflammatory stimulus is applied (e.g. some other infection), the cells remain capable of differentiation and permit HCMV reactivation.
In figures 4 & 5 the authors explore the mechanisms by which HCMV might be affecting innate immune signalling in the GFP- CD34+ cells. Despite reduced IFN transcription, they find the proteins of the cGAS/STING pathway to be increased in GFP- cells. However, in the untreated condition in 4C, the authors do not observe the same as is observed in 4B, at the very least much less convincingly so. How do the authors account for this discrepancy? Potentially the need for an exposure that shows the change following BFA treatment means that it is much more difficult to visualise the difference between untreated mock and GFP-, in which case multiple exposures could be included in the figure. In describing the localisation of p-IRF3 in GFP- cells in response to cGAMP treatment, the authors state that nuclear p-IRF3 was not seen, but clearly from the image it is, it is just that a majority of the p-IRF3 signal lies outside of the nucleus. The authors should update the language to be more precise. The authors also don’t comment on the apparent drop in total p-IRF3 signal in GFP+ cells treated with cGAMP, nor the drop in translocation in these cells. Perhaps the signalling triggered by the exogenous cGAMP in combination with the ongoing increased signalling due to HCMV replication is sufficient to induce a negative feedback loop? Or increased STING activity is more efficiently hijacked by HCMV (see e.g. Costa et al 2024)? The exclusion of p-IRF3 from the nucleus in the GFP- cells is seemingly well established, however. Unfortunately, as acknowledged in the discussion, the authors cannot provide any mechanistic insights into how the virus is able to prevent p-IRF3 translocation in this setting, but this does establish that one mechanism by which HCMV may impact myeloid cell differentiation is by inhibiting type I IFN production by way of impeding p-IRF3 translocation.
The discussion is well balanced and raises a series of questions that naturally follow on from this work. I would be very interested in seeing data on these, in particular the impact of HCMV infection on the ability of cells to differentiate into effective APCs, but this work is beyond the scope of this paper.
The authors need to clarify the source of their cells, as there is no ethics board decision/reference included. If these are truly anonymous donors (say, purchased blood cones produced from blood donation products) this should be clarified.
The quality of the English language is generally very good.
Minor edits:
Line 320: CDP should be CMP
In the later parts of the manuscript, the superscript labelling convention for the GFP+/- cells has been lost.
Specify how MOI was established (line 140-141).
A word is missing from the abstract (line 19), prior to ‘mock infection’. ‘similar to’, ‘akin to’ or something of that nature should be added.
Round 2
Reviewer 1 Report
Comments and Suggestions for Authors
Thanks very much for the authors’ careful revisions. In the revised version, the authors stated that “approximately 56% of GFP-negative cells contained detectable HCMV” (Fig. 1D). It would be helpful if they could describe the calculation methods used to determine these numbers via confocal microscopy.
In Section 3.1, the authors provide additional evidence supporting the existence of HCMV-latently infected cells. However, this evidence mostly confirms whether these cells are infected with HCMV, not whether the GFP-negative cells are specifically infected in a latent state. The GFP-negative population may predominantly consist of uninfected cells. I suggest that the authors culture these purported latently infected cells for an extended period (e.g., more than four weeks) to determine the expression of viral latent genes or proteins, as well as the subsequent reactivation efficiency.
Reviewer 2 Report
Comments and Suggestions for Authors
The revised manuscript appropriately addresses all reviewer comments in my opinion.
Reviewer 3 Report
Comments and Suggestions for Authors
In this revised manuscript, entitled "Latent human cytomegalovirus infection activates the cGAS/STING pathway, but p-IRF3 translocation is limited," Lee WK. et al. investigated HCMV latency established in CD34+ cells isolated from peripheral blood mononuclear cells (PBMC). The RV-TB40-BACKL7-SE-EGFP virus was used to infect peripheral blood-derived CD34+ cells. The GFP+ cells were considered as productively infected (HCMVGFP+ cells), while the GFP- cells were considered as latently infected with HCMV (HCMVGFP- cells). Most of the concerns from the previous review were addressed. However, there are still issues to be addressed. The HCVM-GFP- cells should have more P-IRF3 staining in Ctrl than in Mock Ctrl. Figure 5C and 5D should have a comparison between HCMV-GFP- and Mock. The bracket line on day 5 in Fig. 2D-CDP is too faint. Additionally, the revision did not include comments addressing the reviewers' concerns. No revisions are available for the supplementary data.
